# Revisiting the Power Gains of a Loaded Two-Port: Is There a Missing Element?

Giovanni Ghione  and Marco Pirola *

Department of Electronics and Telecommunications, Politecnico di Torino, Corso Duca degli Abruzzi 24, 10129 Torino, Italy; giovanni.ghione@polito.it
* Correspondence: marco.pirola@polito.it; Tel.: +39-0110904101

**Abstract:** In microwave electronics, the power gains of a linear two-port are customarily defined as the ratio of an output port and input port power, where such powers are intended either as operational or as available. Two input and two output powers are thus introduced, with four possible combinations of output/input power ratios, but only three are practically exploited, the well-known operational power gain, available power gain, and transducer power gain. In the present paper, we provide a comprehensive review of gain definitions (including the less commonly exploited added-power gains) and finally consider the missing fourth element (defined as the ratio of the output available power and of the input operational power), derive a few mathematical properties of it, both in the general and in the unilateral case, and ultimately justify the reason why this fourth gain $G_4$ which, following the suggestion of an anonymous reviewer, we will call apparent power gain, $G_{app}$, has little interest in the optimization of the power transfer between the generator and the load. Nevertheless, the definition and analysis of $G_{app}$, besides being formally useful to complete the gain family, may yield a deeper insight into the very nature of power transfer optimization in a loaded two-port.

**Keywords:** circuit stability; input and output stability; linear circuits; scattering parameters; stability; stability criteria; two-port circuits



## 1. Introduction: Reviewing Two-Port Gains

Consider a two-port of scattering matrix $S$, obtained through circuit analysis, CAD simulation, or experimental characterization [1], connected at port 1 (the input port) to a generator of internal reflectance $\Gamma_G$ and loaded at port 2 (the output port) by a reflectance $\Gamma_L$; see Figure 1. All textbooks on microwave electronics agree on defining four active powers (or power spectral densities), two relevant to the input port and two to the output port, albeit with slightly different names, see, e.g., [2] (Section 3.2), [3] (Section 10.5), [4] (Section 6.3), as follows:

- The input power $P_{in}$, i.e., the actual power input to the network;
- The input available power $P_{av,in}$, i.e., the available power of the input generator;
- The power actually delivered to the load, $P_L$;
- The output available power $P_{av,L}$, i.e., the available power from port 2.

The available power of a generator is defined as the maximum power that the generator can deliver when it is connected to a conjugately matched load.

A power gain between port 2 and port 1 can be in principle defined as the ratio of any of the output powers vs. any of the input powers. According to the Friis definition [5], "The gain of the network is defined as the ratio of the available signal power at the output terminals of the network to the available signal power at the output terminals of the signal generator", i.e., $G = P_{av,L}/P_{av,in}$ (This definition is consistent with the fact that in noise analysis all powers considered are available powers). Two years later, Roberts [6]

wrote: "We shall use the symbol for gain proposed by Friis, but shall modify his definition slightly. The gain is defined here as the ratio of the power delivered to the load impedance connected at the output terminals to the power available from the generator connected at the input terminals", i.e., $G = P_L/P_{av,in}$ (In fact, Friis [5] already noticed that his definition was "an unusual definition of gain since the gain of an amplifier is generally defined as the ratio of its output and input powers"). Finally, in 1954, Mason [7] mentioned a "source-to-load power gain" to be interpreted probably as $G = P_L/P_{in}$; later, in 1959, Venkateswaran and Boothroyd explicitly introduced this term as the "operating power gain" [8]. Currently, the three gains introduced are denoted by different symbols, as follows (see, e.g., [2] (Section 3.2), [3] (Section 10.5), [4] (Section 6.3)):

$$G_{op} = \frac{P_L}{P_{in}} \text{ the operational (operative, operating) power gain}$$

$$G_{av} = \frac{P_{av,L}}{P_{av,in}} \text{ the available power gain}$$

$$G_t = \frac{P_L}{P_{av,in}} \text{ the transducer (power) gain}$$

For the sake of completeness, we recall here the expression of the gains as a function of the two-port scattering parameters and of the load and generator reflectances, see, e.g., [4] (Sections 6.3.2, 6.3.3, 6.3.4). (Other expressions, probably more elegant, can be derived in terms of impedance, admittance, or hybrid parameters):

$$G_{op}(\Gamma_L) = |S_{21}|^2 \frac{1 - |\Gamma_L|^2}{|1 - S_{22}\Gamma_L|^2 - |S_{11} - \Delta_S\Gamma_L|^2}$$

$$G_{av}(\Gamma_G) = |S_{21}|^2 \frac{1 - |\Gamma_G|^2}{|1 - S_{11}\Gamma_G|^2 - |S_{22} - \Delta_S\Gamma_G|^2}$$

$$G_t(\Gamma_L, \Gamma_G) = |S_{21}|^2 \times$$

$$\times \frac{\left(1 - |\Gamma_L|^2\right)\left(1 - |\Gamma_G|^2\right)}{|(1 - S_{11}\Gamma_G)(1 - S_{22}\Gamma_L) - S_{12}S_{21}\Gamma_G\Gamma_L|^2}$$

where $\Delta_S = S_{11}S_{22} - S_{12}S_{21}$ is the determinant of the scattering matrix.

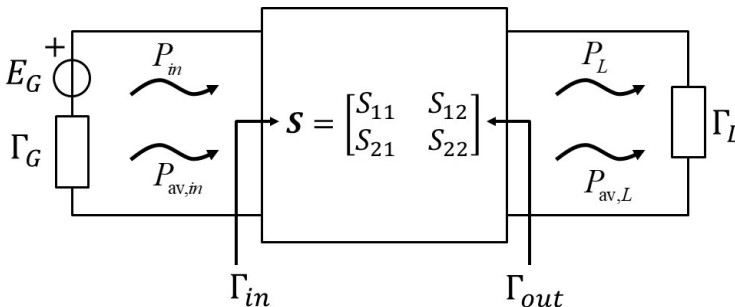

**Figure 1.** Two-port with scattering matrix $S$ loaded at the input by a generator of reflectance $\Gamma_G$ and at the output by a load of reflectance $\Gamma_L$. The input and output reflectances of the loaded two-port are denoted by $\Gamma_{in}$ and $\Gamma_{out}$, respectively. Input and output total active powers at the input and output are also indicated.

If the two-port is unconditionally stable, the power transfer between generator and load can be maximized by simultaneous conjugate matching at the input and output ports, such as the input (output) two-port reflectance $\Gamma_{in}$ ($\Gamma_{out}$) equaling the complex conjugate of the generator (load) reflectance, $\Gamma_G^*$ ($\Gamma_L^*$). This corresponds to a maximum gain value that is obviously the same for all gains, since maximum power transfer implies that $P_{in} = P_{av,in}$ and that $P_L = P_{av,L}$; for historical reasons, the maximum gain is often referred to as max-

imum available gain or MAG. However, notice that since the transducer gain depends on both the load and the generator reflectances, its maximization directly yields maximum power transfer between generator and load. Conversely, input (output) conjugate matching is additionally required to obtain the maximum power transfer when $G_{op}$ is maximized vs. $\Gamma_L$ ($G_{av}$ vs. $\Gamma_G$). On the other hand, if the two-port is not unconditionally stable, simultaneous conjugate matching cannot be implemented, although a lower bound exists to the input and output mismatch depending on the Rollet stability factor $K$ [9], as recently demonstrated by the present authors in [10].

For the sake of completeness, we also report the gains in the so-called unilateral case, i.e., for a device with $S_{12} = 0$. The past popularity of the unilateral approach to amplifier design is motivated by a number of facts. First, most active devices are almost unilateral (i.e., $S_{12}$ is small in magnitude). Second, unilateral design is easy and could be carried out by hand before the advent of CAD tools. Finally, devices could be made unilateral (at least in theory) by proper circuit design, see e.g., [7]. For these reasons, gains are sometimes defined with reference to a device that is or is made unilateral, with the notation $G_{op}^u$, $G_{av}^u$, $G_t^u$. The MAG of a unilateral device is denoted as MUG, and the unilateral operational, available, and transducer power gains read:

$$G_{op}^u(\Gamma_L) = |S_{21}|^2 \frac{1 - |\Gamma_L|^2}{\left(1 - |S_{11}|^2\right)|1 - S_{22}\Gamma_L|^2}$$

$$G_{av}^u(\Gamma_G) = |S_{21}|^2 \frac{1 - |\Gamma_G|^2}{\left(1 - |S_{22}|^2\right)|1 - S_{11}\Gamma_G|^2}$$

$$G_t^u(\Gamma_L, \Gamma_G) = |S_{21}|^2 \frac{\left(1 - |\Gamma_L|^2\right)\left(1 - |\Gamma_G|^2\right)}{|(1 - S_{11}\Gamma_G)(1 - S_{22}\Gamma_L)|^2}$$

A further possibility in defining two-port gain involves the concept of added power, i.e., the difference between the power delivered to the load and the input power, $P_L - P_{in}$. In large-signal operation, the added-power concept is the basis for the power-added efficiency parameter (see, e.g., [4] (Section 8.2.4)). A definition and optimization of the two-port gain based on the added power has been proposed in the past by Kotzebue [11–13] and recently exploited in [14]. To introduce the subject, let us consider as a possible gain parameter, $G_{add}$ (additional power gain), the ratio between the added power and the input power:

$$G_{add} = \frac{P_L - P_{in}}{P_{in}} = \frac{P_L}{P_{in}} - 1 = G_{op} - 1 \tag{1}$$

Alternative mathematical definitions of the additional power gain could involve the available input or load powers instead of the operational powers, or a combination of them. Those definitions however have never been proposed so far and seem to be of little use since the only physically meaningful definition is the one in (1). Optimization of $G_{add}$ is apparently straightforward since simultaneous conjugate matching results in $G_{op} = \text{MAG}$. The maximum value of $G_{add}$ will be therefore $\text{MAG} - 1$. Interestingly enough, the so-called maximally efficient gain defined in [12,13] does not correspond to $\text{MAG} - 1$, nor to simultaneous conjugate matching. Let us try to summarize the optimization concept proposed in the above papers. In [13], a two-port loading condition is investigated leading to the optimization of $P_L - P_{in}$ with constant square magnitude of the input voltage, i.e., defining:

$$P_L - P_{in} = g|V_1|^2$$

The maximum of $g$, here called $g_{ME}$ (where the subscript stands for maximally efficient, see [12]), is defined in [13] (Equation (6)) in terms of the two-port admittance parameters.

This maximum is achieved with the two-port load defined in [13] (Equation (7)); the generator is power-matched. The equations are reported here for completeness:

$$g_{\text{ME}} = \frac{|Y_{21}|^2 + |Y_{12}|^2 + 2\text{Re}(Y_{21}Y_{12}) - 4\text{Re}(Y_{11})\text{Re}(Y_{22})}{4\text{Re}(Y_{22})} \tag{2}$$

$$Y_{L,\text{opt}} = \frac{2Y_{21}\text{Re}(Y_{22})}{Y_{21} + Y_{12}^*} - Y_{22} \tag{3}$$

$$Y_{G,\text{opt}} = Y_{in}^* \tag{4}$$

The parameter $g$ (and its maximum $g_{\text{ME}}$) clearly is a transconductance rather than a power gain, and the input power $P_{in}$ will depend not only on $|V_1|^2$, but also on the input reflectance or immitance of the two-port. In [12], the optimum parameter $g_{\text{ME}}$ is reformulated, albeit in a slightly different form, as the "maximally efficient gain", $G_{\text{ME}}$, defined, somewhat obscurely, as "the power gain which maximizes the 2-port added power for a given value of the input port independent variable". The parameter is defined as:

$$G_{\text{ME}} = \frac{|Y_{21}|^2 - |Y_{12}|^2}{4\text{Re}(Y_{11})\text{Re}(Y_{22}) - 2\text{Re}(Y_{21}Y_{12}) - 2|Y_{12}|^2} \tag{5}$$

with the optimum load and generator admittance defined again as in (3) and (4). The same equations are reported in [14] (Equations (1)–(3)), correcting a sign misprint in [12] (Equation (3)). The maximally efficient gain can also be reformulated in terms of S-parameters as:

$$G_{\text{ME}} = \frac{|S_{21}/S_{12}|^2 - 1}{2(K|S_{21}/S_{12}| - 1)}$$

(see [12] (Equation (4))) where $K$ is the Rollet stability factor. It is important to stress that the maximally efficient gain $G_{\text{ME}}$ does not actually correspond to the maximum additional power gain $G_{\text{add,max}} = \text{MAG} - 1$, but rather to the operational power gain $G_{\text{op}}$ evaluated in the optimum load condition (3).

The relationship between the parameters $g_{\text{ME}}$ (2) and $G_{\text{ME}}$ (5) can be easily derived taking into account that $P_{in} = |V_1|^2\text{Re}(Y_{in})$, where $Y_{in}$ is the two-port input admittance. We therefore have:

$$(P_L - P_{in})_{\text{max}} = g_{\text{ME}}|V_1|^2 = \frac{g_{\text{ME}}}{\text{Re}(Y_{in})}P_{in}$$

i.e.,

$$\left(\frac{P_L - P_{in}}{P_{in}}\right)_{\text{max}} = \left(\frac{P_L}{P_{in}}\right)_{\text{max}} - 1 = G_{\text{ME}} - 1 = \frac{g_{\text{ME}}}{\text{Re}(Y_{in})} \rightarrow G_{\text{ME}} = \frac{g_{\text{ME}}}{\text{Re}(Y_{in})} + 1.$$

A number of advantages related to the choice of this optimization paradigm are reported in [11–13] (this choice would be a batter compromise in terms of the amplifier large-signal saturation behavior) and in [14] (where the technique is exploited to optimize gain-boosted amplifiers working close to the maximum oscillation frequency). Furthermore, application of the maximally efficient gain concept to high-power oscillator design was first proposed in [15]. (In [15], the $G_{\text{ME}}$ concept is formulated as "the power gain which maximizes the two-port added power", and expressions are provided for the optimum load and generator reflectances).

## 2. Is a Fourth Gain Missing?

A fourth possibility for defining a gain indeed exists, as follows:

$$G_{\text{app}} = \frac{P_{\text{av},L}}{P_{in}} \tag{6}$$

In this section, we intend to explore the properties of the apparent power gain $G_{\text{app}}$, anticipating that this parameter has some interesting mathematical properties but is of little use in circuit design, since, as the name itself suggests, it does not generally correspond to a physically realizable circuit configuration, as discussed in detail in Section 3. First of all, we point out that the apparent gain is not independent from the other ones; indeed:

$$G_{\text{app}} = \frac{P_{\text{av},L}}{P_{in}} = \frac{P_{\text{av},L}}{P_{in}} \frac{P_{\text{av},in}}{P_{\text{av},in}} \frac{P_L}{P_L} = \tag{7}$$

$$= \frac{P_{\text{av},L}}{P_{\text{av},in}} \frac{P_L}{P_{in}} \frac{P_{\text{av},in}}{P_L} = \frac{G_{\text{av}} G_{\text{op}}}{G_{\text{t}}},$$

or:

$$G_{\text{app}} G_{\text{t}} = G_{\text{av}} G_{\text{op}}$$

Moreover, since:

$$P_{\text{av},in} \geq P_{in}, \quad P_{\text{av},L} \geq P_L$$

this implies the following inequalities:

$$\frac{P_L}{P_{in}} \geq \frac{P_L}{P_{\text{av},in}} \rightarrow G_{\text{op}} \geq G_{\text{t}}$$

$$\frac{P_{\text{av},L}}{P_{\text{av},in}} \geq \frac{P_L}{P_{\text{av},in}} \rightarrow G_{\text{av}} \geq G_{\text{t}}$$

$$\frac{P_{\text{av},L}}{P_{in}} \geq \frac{P_L}{P_{in}} \rightarrow G_{\text{app}} \geq G_{\text{op}}$$

$$\frac{P_{\text{av},L}}{P_{in}} \geq \frac{P_{\text{av},L}}{P_{\text{av},in}} \rightarrow G_{\text{app}} \geq G_{\text{av}}$$

i.e.,

$$G_{\text{app}} \geq \left\{ G_{\text{av}}, G_{\text{op}} \right\} \geq G_{\text{t}}.$$

The only condition corresponding to the equal sign is when $G_{\text{app}} = G_{\text{t}} = G_{\text{av}} = G_{\text{op}} = \text{MAG}$.

In general, $G_{\text{app}}$ can be, according to the loading conditions, larger or smaller than MAG. The reason for this behavior is that the output available power becomes the output power in case of output matching, but setting the load reflectance accordingly changes in turn the input power. In other words, the apparent gain does not refer to a physically realizable circuit configuration unless conjugate matching is imposed at both ports.

A schematic illustration of this is depicted in Figure 2, where the maximization of power transfer between the generator $G$ and the load $L$ through reactive matching sections placed at the two-port input and output, respectively, is depicted as a block scheme. Simultaneous conjugate matching implies that the generator available power is the two-port input power and that the power delivered to the load is the two-port output available power. As shown by the grey shadowed box in Figure 2a, this directly corresponds to the maximization of the transducer gain. Maximizing the operational gain (see the grey shadowed box in Figure 2b) is a necessary but not sufficient condition for maximum power transfer since it does not automatically include the input conjugate matching. The same remark applies to the maximization of the available power gain (see the grey shadowed box in Figure 2c); this time, the output conjugate matching is not included and has to be separately implemented in order to maximize the power transfer. Finally, as shown in Figure 2d, regardless of which condition we may impose on the apparent gain, this does not imply either the input or the output conjugate matching. However, if the input and output reactive matching sections are designed so as to obtain input and output simultaneous conjugate matching (that is, the necessary and sufficient condition for maximum power transfer), the value of the apparent gain will consistently correspond to the MAG.

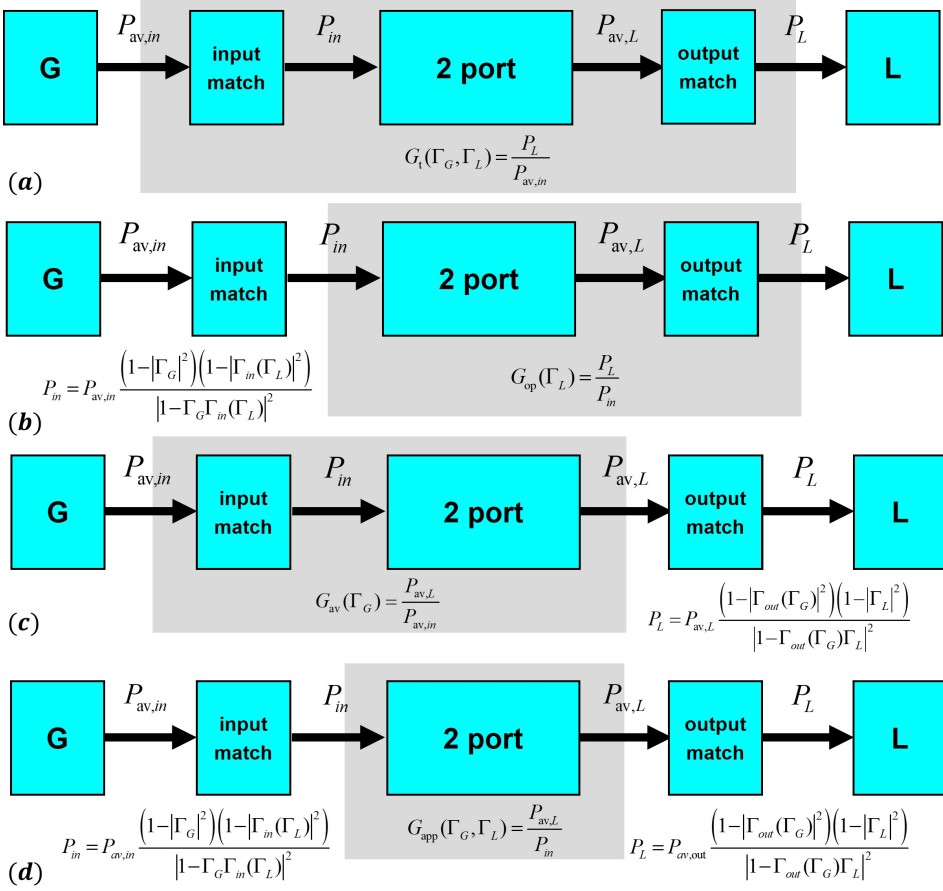

**Figure 2.** Block scheme of the power transfer in a loaded two-port. The input and output reactive matching sections are meant to provide simultaneous input/output conjugate matching, thus allowing for maximum power transfer between the generator and the load. In case (**a**), the maximization of the transducer gain is enough to allow for maximum power transfer, since it includes input and output matching. In cases (**b,c**), optimization of the operational or available power gain is a necessary condition for maximum power transfer, but not a sufficient one since, additionally, the input or output conjugate matching has to be implemented to this aim. Case (**d**), concerning $G_{app}$, suggests that this parameter is somewhat irrelevant to power transfer maximization, since the implementation of the input and output conjugate matchings is anyway required. See however the text for further comments on how $G_{app}$ can be exploited in the implementation of simultaneous conjugate matchings.

We further observe that, from this scheme, it appears that an alternative to maximize the input–output power transfer by maximizing $G_t$ is to optimize $G_{av}$ and $G_{op}$ together, e.g., to optimize their product $G_{av}G_{op}$, or a monotonically increasing function of it.

As expected, $G_{app}$ depends on both the load and the generator reflectances. Its explicit expression is:

$$G_{app}(\Gamma_L, \Gamma_G) = \frac{|S_{21}|^2}{|1 - S_{22}\Gamma_L|^2 - |S_{11} - \Delta_S\Gamma_L|^2} \times$$

$$\times \frac{|1 - S_{11}\Gamma_G - S_{22}\Gamma_L + \Delta_S\Gamma_G\Gamma_L|^2}{|1 - S_{11}\Gamma_G|^2 - |S_{22} - \Delta_S\Gamma_G|^2}$$

Taking into account that:

$$\Gamma_{in}(\Gamma_L) = \frac{S_{11} - \Delta_S\Gamma_L}{1 - S_{22}\Gamma_L}$$

$$\Gamma_{out}(\Gamma_G) = \frac{S_{22} - \Delta_S\Gamma_G}{1 - S_{11}\Gamma_G}$$

we can also write:

$$G_{\text{app}} = \frac{|S_{21}|^2}{\left(1 - |\Gamma_{in}(\Gamma_L)|^2\right)\left(1 - |\Gamma_{out}(\Gamma_G)|^2\right)} \times$$
$$\times \left|1 - \frac{S_{12}S_{21}\Gamma_G\Gamma_L}{(1 - S_{11}\Gamma_G)(1 - S_{22}\Gamma_L)}\right|^2 \approx$$
$$\approx \frac{|S_{21}|^2}{\left(1 - |\Gamma_{in}(\Gamma_L)|^2\right)\left(1 - |\Gamma_{out}(\Gamma_G)|^2\right)} \quad (8)$$

Equation (8) is a partial unilateral approximation of $G_{\text{app}}$, i.e., the term $|\cdot|^2$ has been approximated to unity while the input and output reflectances have been computed with the actual value of $S_{12}$. For potentially unstable devices, $G_{\text{app}}$ diverges when $|\Gamma_{in}(\Gamma_L)| \to 1$ or $|\Gamma_{out}(\Gamma_G)| \to 1$, which defines again the customary load/generator stability circles. Finally, a somewhat surprising result is obtained for a completely unilateral device ($S_{12} = 0$). In such a case, $\Gamma_{in} = S_{11}$ and $\Gamma_{out} = S_{22}$, and therefore:

$$G_{\text{app}}^u = \frac{|S_{21}|^2}{\left(1 - |S_{11}|^2\right)\left(1 - |S_{22}|^2\right)} \equiv \text{MUG} \quad (9)$$

In other words, for a unilateral device $G_{\text{app}}$ is constant, independent on the load conditions, and equal to the maximum unilateral gain. A consequence of (9) is that, for a unilateral device, the following equality holds independent of $\Gamma_G$ and $\Gamma_L$:

$$\frac{G_{\text{av}}^u G_{\text{op}}^u}{G_{\text{t}}^u} = \text{MUG}.$$

## 3. Examples and Discussion

In this section, we further explore the properties of $G_{\text{app}}$ on the basis of a case study. A first issue to be discussed is whether this parameter may be of any use in the numerical optimization of the narrowband two-port gain through proper input and output matching sections.

Let us suppose first that the input matching section is designed so as to obtain $G_{\text{av}} = \text{MAG}$; in this case:

$$G_{\text{app}} = \text{MAG} \cdot \frac{P_{\text{av},in}}{P_{in}} \geq \text{MAG};$$

thus, optimization of $G_{\text{app}}$ with goal $G_{\text{app}} = \text{MAG}$ would lead to simultaneous conjugate matching at the two ports. In practice, this would correspond to an optimization process with two simultaneous goals, maximum $G_{\text{av}}$ and minimum $G_{\text{app}}$ (or $G_{\text{app}} = \text{MAG}$). As a dual case, suppose that the output matching section is designed so as to obtain $G_{\text{op}} = \text{MAG}$; in this case:

$$G_{\text{app}} = \text{MAG} \cdot \frac{P_{\text{av},L}}{P_L} \geq \text{MAG};$$

thus, optimization of $G_{\text{app}}$ with goal $G_{\text{app}} = \text{MAG}$ would lead to simultaneous conjugate matching at the two ports. This would again correspond to an optimization process with two simultaneous goals, maximum $G_{\text{op}}$ and minimum $G_{\text{app}}$ (or $G_{\text{app}} = \text{MAG}$).

On the other hand, optimizing $G_{\text{app}}$ only with goal $G_{\text{app}} = \text{MAG}$ does not lead to a unique solution for the input and output reflectances which, moreover, do not necessarily correspond to the optimum ones. This non-uniqueness is readily suggested by the following

example. Suppose $\Gamma_L = 0 \neq \Gamma_{L,\text{op}}$ where $\Gamma_{L,\text{op}}$ is the load reflectance corresponding to simultaneous conjugate matching; we have, imposing $G_{\text{app}} = \text{MAG}$:

$$G_{\text{app}}(0, \Gamma_G) = \frac{|S_{21}|^2}{\left(1 - |S_{11}|^2\right)\left(1 - |\Gamma_{out}(\Gamma_G)|^2\right)} \equiv$$

$$= \frac{G_{\text{op}}(0)}{1 - |\Gamma_{out}(\Gamma_G)|^2} = \text{MAG}$$

This condition implies:

$$|\Gamma_{out}(\Gamma_G)| = \sqrt{1 - \frac{G_{\text{op}}(0)}{\text{MAG}}} = \text{const.} \tag{10}$$

Since the transformation $\Gamma_G \rightarrow \Gamma_{out}$ is a bilinear transformation (see e.g., [2]) that maps circles into circles, the circles $|\Gamma_{out}(\Gamma_G)| = \text{const.}$ generally correspond to a circle (or circle arc) in $\Gamma_G$ plane, i.e., many $\Gamma_G$ values satisfy the constraint in (10). This circle degenerates into a point only if $\Gamma_L = \Gamma_{L,\text{op}}$; for example, if $\Gamma_{L,\text{op}} = 0$ we obtain from (10) that $|\Gamma_{out}(\Gamma_G)| = 0 \rightarrow \Gamma_{out}(\Gamma_G) = 0 = \Gamma_{L,\text{op}}^*$, implying a single value of $\Gamma_G = \Gamma_{G,\text{op}}$. A similar situation holds for any given generator or load different from the optimum one.

From the very definition of $G_{\text{app}}$ reported in (6), we also see that the cases of a reactive load or generator reflectance do not lead to any particularly significant behavior of this parameter, since $P_{\text{av},L}$ is unaffected by the load termination and $P_{in}$ is regular also in the presence of a generator with reactive reflectance. In general, for an unconditionally stable two-port, $G_{\text{app}}$ will span between a minimum (generally nonzero) and maximum value, as suggested by the following numerical example. Assume the following constant scattering matrix:

$$S_{11} = 0.61 \exp(j\, 0.916\,67\pi)$$
$$S_{21} = 3.72 \exp(j\, 0.327\,78\pi)$$
$$S_{12} = 0.05 \exp(j\, 0.233\,33\pi)$$
$$S_{22} = 0.45 \exp(-j\, 0.266\,67\pi)$$

The stability parameters are $K = 1.1752$ and $|\Delta_S| = 0.1086$; the two-port is therefore unconditionally stable with optimum load and source reflectances:

$$\Gamma_{L,\text{op}} = 0.7495 \exp(j\, 0.2920\pi)$$
$$\Gamma_{G,\text{op}} = 0.8179 \exp(j\, 1.0963\pi)$$

and maximum available gain MAG $= 41.5032$. As a first example, we set $\Gamma_L = \Gamma_{L,\text{op}}$ and scan the $\Gamma_G$ plane (considering passive loads only), then we plot the surface $G_{\text{app}}(|\Gamma_G|, \phi_G/\pi)$ where $\phi_G$ is the phase of $\Gamma_G$. The result is shown in Figure 3; $G_{\text{app}}$ clearly exhibits a unique minimum in the $\Gamma_G$ plane corresponding to the MAG. Dually, we set $\Gamma_G = \Gamma_{G,\text{op}}$ and scan the $\Gamma_L$ plane (considering passive loads only), then we plot the surface $G_{\text{app}}(|\Gamma_L|, \phi_L/\pi)$ where $\phi_L$ is the phase of $\Gamma_L$. The result is shown in Figure 4; again, a unique minimum exists in correspondence with $\Gamma_{L,\text{op}}$.

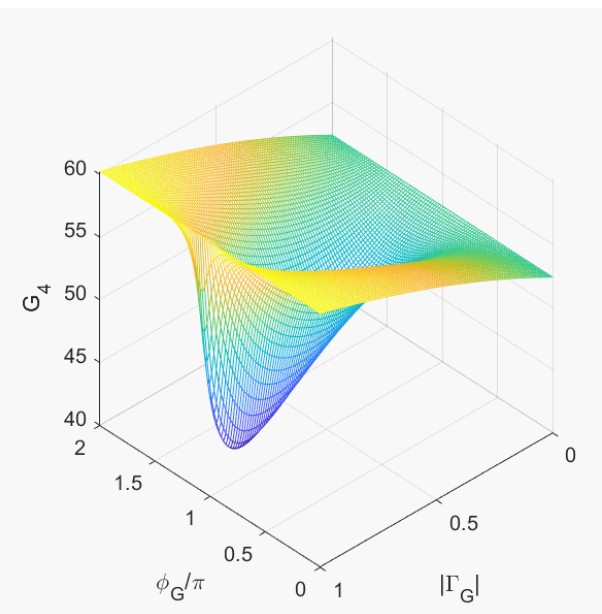

**Figure 3.** Surface plot of $G_4 \equiv G_{\text{app}}$ as a function of the generator reflectance (magnitude and phase); the unique minimum is located at $\Gamma_{G,\text{op}}$ and its value corresponds to the two-port MAG.

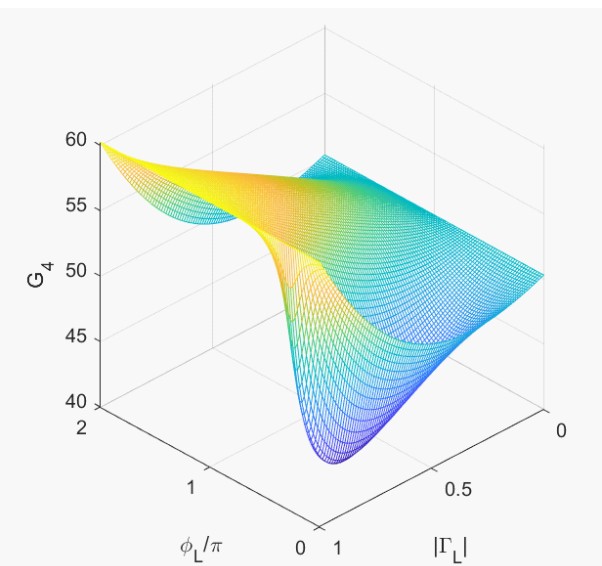

**Figure 4.** Surface plot of $G_4 \equiv G_{\text{app}}$ as a function of the load reflectance (magnitude and phase); the unique minimum is located at $\Gamma_{L,\text{op}}$ and its value corresponds to the two-port MAG.

Exploring the behavior of $G_{\text{app}}$ in the $(\Gamma_L, \Gamma_G)$ four-dimensional space is more complex. To provide some hint, we perform a random sampling of the $(\Gamma_L, \Gamma_G)$ space, sort the $G_{\text{app}}$ samples in increasing order, and generate a plot of $G_{\text{app}}$ vs. the sample order together with a scatter plot of the corresponding values of $G_{\text{av}}$, $G_{\text{op}}$, $G_{\text{t}}$. The $G_{\text{app}}$ plot together with the unilateral approximation in (8) is shown in Figure 5. The minimum and maximum $G_{\text{app}}$ samples depend of course on the number $N$ of samples considered; with $N = 100,000$, we have $G_{\text{app,min}} \approx 17.1661$ and $G_{\text{app,max}} \approx 189.8973$.

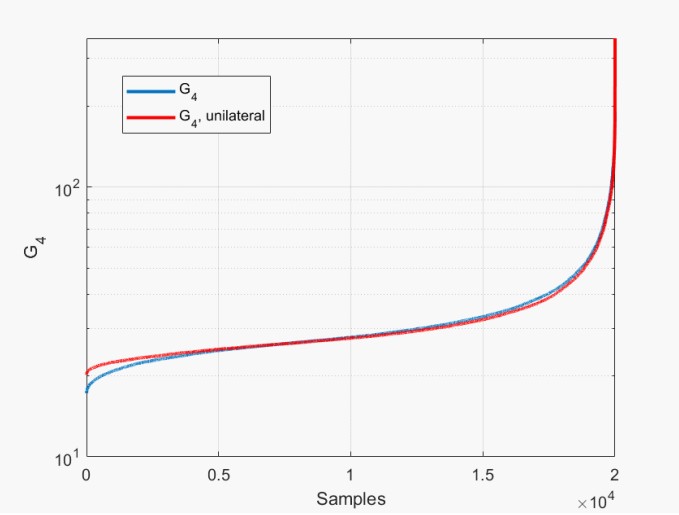

**Figure 5.** Plot of $G_4 \equiv G_{\text{app}}$ samples in increasing order and of the unilateral approximation in (8) vs. sample order (20,000 samples wereused here).

Finally, Figure 6 reports a scatter plot of $G_{\text{av}}$, $G_{\text{op}}$, $G_t$ (the samples have been ordered with increasing value of $G_{\text{app}}$). We remark again that $G_{\text{app}}$ can be lower or larger than the MAG and that close to the MAG value the samples of $G_{\text{av}}$, $G_{\text{op}}$, and $G_t$ exhibit a dense clustering, thus confirming that the condition $G_{\text{app}} = \text{MAG}$ does not necessarily correspond to simultaneous conjugate matching. From the scatter plot, it appears that the maximum and minimum of $G_{\text{app}}$ are associated with load and generator reflectances for which $G_{\text{av}}$, $G_{\text{op}}$, and $G_t$ vanish, i.e., with reactive reflectances. A Monte Carlo analysis carried out on $N =$100,000 samples indeed suggests that $G_{\text{app,min}}$ is associated with the load and generator impedances with both being reactive, while $G_{\text{app,max}}$ is associated with just one reactive termination; however, reactivity alone does not necessarily imply extremum values, since the phase of the reactive termination also has to be properly chosen.

The maximum ratio $P_{\text{av},L} / P_{\text{av},in}$ can be derived numerically by maximizing the numerator and minimizing the denominator. We have:

$$
\begin{aligned}
P_{\text{av},L} &= G_{\text{av}}(\Gamma_G) P_{\text{av},in} = \\
&= \frac{|S_{21}|^2 \left(1 - |\Gamma_G|^2\right)}{|1 - S_{11}\Gamma_G|^2 - |S_{22} - \Delta_S\Gamma_G|^2} \times \frac{|b_0|^2}{1 - |\Gamma_G|^2} = \\
&= |S_{21}|^2 \frac{|b_0|^2}{|1 - S_{11}\Gamma_G|^2 - |S_{22} - \Delta_S\Gamma_G|^2}
\end{aligned}
$$

where $b_0$ is the amplitude of the input forward wave generator connected to port 1 of the two-port. This maximum $P_{\text{av},L,\text{max}}$ is obtained for a specific value of the generator reflectance, $\Gamma_G = \Gamma_{G,\text{max}}$, i.e.,:

$$
P_{\text{av},L,\text{max}} = |S_{21}|^2 \frac{|b_0|^2}{|1 - S_{11}\Gamma_{G,\text{max}}|^2 - |S_{22} - \Delta_S\Gamma_{G,\text{max}}|^2}
$$

The input power can be now expressed as:

$$
P_{in} = |b_0|^2 \frac{1 - |\Gamma_{in}(\Gamma_L)|^2}{|1 - \Gamma_{G,\text{max}}\Gamma_{in}(\Gamma_L)|^2}
$$

which is minimum for a certain value of $\Gamma_L = \Gamma_{L,\text{min}}$, i.e.,:

$$
P_{in,\text{min}} = |b_0|^2 \frac{1 - |\Gamma_{in}(\Gamma_{L,\text{min}})|^2}{|1 - \Gamma_{G,\text{max}}\Gamma_{in}(\Gamma_{L,\text{min}})|^2}
$$

Thus:

$$G_{\text{app,max}} = \frac{P_{\text{av},L,\text{max}}}{P_{in,\text{min}}} =$$

$$= \frac{|S_{21}|^2}{|1 - S_{11}\Gamma_{G,\text{max}}|^2 - |S_{22} - \Delta_S\Gamma_{G,\text{max}}|^2} \times$$

$$\times \frac{|1 - \Gamma_{G,\text{max}}\Gamma_{in}(\Gamma_{L,\text{min}})|^2}{1 - |\Gamma_{in}(\Gamma_{L,\text{min}})|^2}$$

Conversely, the minimum ratio $P_{\text{av},L}/P_{in}$ can be obtained as:

$$G_{\text{app,min}} = \frac{P_{\text{av},L,\text{min}}}{P_{in,\text{max}}}$$

Again, the value of $P_{\text{av},L,\text{min}}$ is obtained for a specific value of the generator reflectance, $\Gamma_G = \Gamma_{G,\text{min}}$, i.e.,:

$$P_{\text{av},L,\text{min}} = |S_{21}|^2 \frac{|b_0|^2}{|1 - S_{11}\Gamma_{G,\text{min}}|^2 - |S_{22} - \Delta_S\Gamma_{G,\text{min}}|^2}$$

while $P_{in}$ is maximum for a certain value of $\Gamma_L = \Gamma_{L,\text{max}}$, i.e.,:

$$P_{in,\text{max}} = |b_0|^2 \frac{1 - |\Gamma_{in}(\Gamma_{L,\text{max}})|^2}{|1 - \Gamma_{G,\text{min}}\Gamma_{in}(\Gamma_{L,\text{max}})|^2}$$

Thus:

$$G_{\text{app,min}} = \frac{P_{\text{av},L,\text{min}}}{P_{in,\text{max}}} =$$

$$= \frac{|S_{21}|^2}{|1 - S_{11}\Gamma_{G,\text{min}}|^2 - |S_{22} - \Delta_S\Gamma_{G,\text{min}}|^2} \times$$

$$\times \frac{|1 - \Gamma_{G,\text{min}}\Gamma_{in}(\Gamma_{L,\text{max}})|^2}{1 - |\Gamma_{in}(\Gamma_{L,\text{max}})|^2}$$

With the scattering matrix exploited in the example, we have:

$$G_{\text{app,max}} = 212.2846$$
$$\Gamma_{G,\text{max}} = -0.9546 - j0.2980 = \exp(-j2.8390)$$
$$\Gamma_{L,\text{min}} = -0.1267 - j0.9919 = \exp(-j1.6979)$$

and:

$$G_{\text{app,min}} = 17.1016$$
$$\Gamma_{G,\text{min}} = 0.9547 + j0.2977 = \exp(j0.3023)$$
$$\Gamma_{L,\text{max}} = -0.1205 - j0.9927 = \exp(-j1.6916)$$

confirming that the terminations corresponding to the maximum and minimum of $G_{\text{app}}$ are reactive. The values obtained by direct search compare well with the previous Monte Carlo analysis.

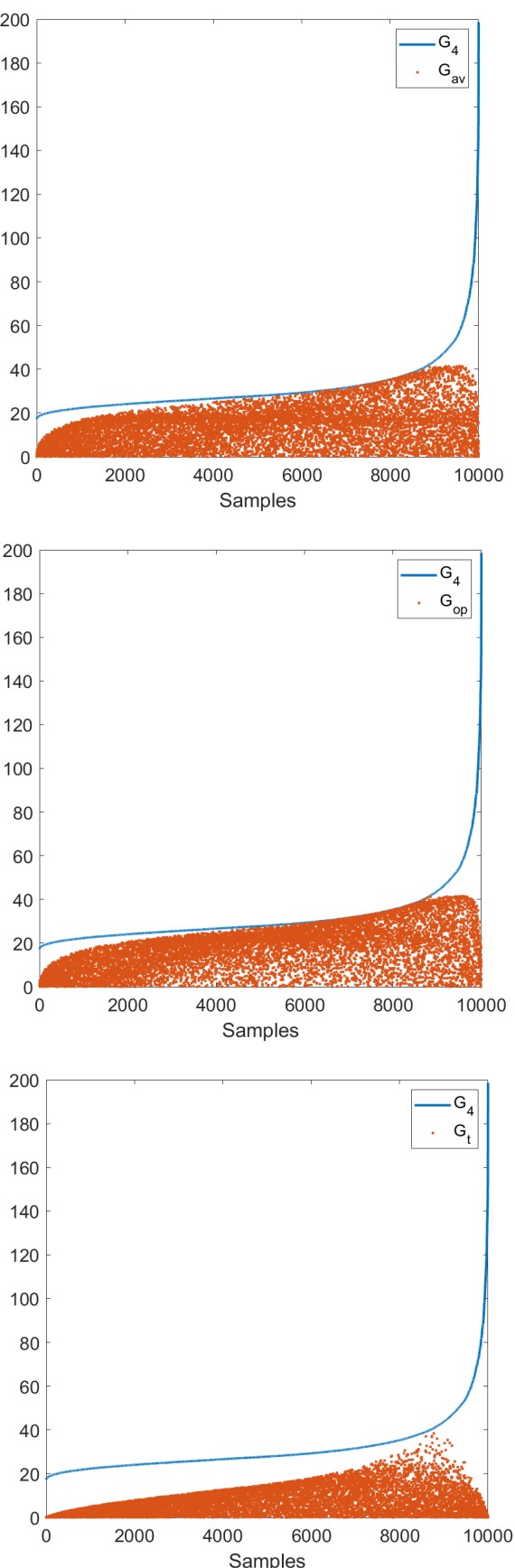

**Figure 6.** Plot of $G_4 \equiv G_{app}$ vs. the sample order together with a scatter plot of the corresponding values of $G_{av}$, $G_{op}$, $G_t$ (top, center, bottom). The Monte Carlo analysis was made with $N = 10,000$ samples.

Finally, it is essential to emphasize the somewhat artificial nature of $G_{\text{app}}$ through a few key points. Let us reconsider the maximum value of $G_{\text{app}}$ and the related maximum available power, $P_{\text{av},L,\text{max}}$, that, as shown in the previous example, occurs by connecting at port 1 a generator with reactive internal impedance. (Notice that while such an ideal generator introduces no anomaly, its available power tends to infinity). The real issue is whether $P_{\text{av},L,\text{max}}$ can be actually delivered to a load. The only possible way to do this is by conjugately matching the two-port at the output. But this obviously corresponds to the maximum operational gain $G_{\text{op}} = \text{MAG}$ (while the available gain and the transducer gain are zero, since the source available power is infinite). Unfortunately, such an output matching is inconsistent with the condition leading to $P_{in,\text{min}}$, which implies a reactive load at port 2 and therefore zero power on the load. This ultimately stresses that the input and output loading conditions corresponding to the maximum $G_{\text{app}}$ are totally inconsistent with the purpose of maximum power transfer. This would anyway require conjugate matching at both ports, thus leading to $G_{\text{op}} = \text{MAG} < G_{\text{app},\text{max}}$. In conclusion, maximizing $G_{\text{app}}$ does not align with a generator and load condition compatible with maximum power transfer (and similar comments may be made about minimizing $G_{\text{app}}$). Nevertheless, understanding the artificial nature of $G_{\text{app}}$ holds, in our opinion, the merit of providing a deeper insight into the fundamental nature of power transfer optimization within a loaded two-port system.

## 4. Conclusions

In this contribution, we have revisited the well-known topic of power transfer maximization in a linear, unconditionally stable two-port through gain optimization. After reviewing the standard input–output gain definitions (i.e., the well-known operational, available, and transducer power gains) and those less commonly used in linear design based on the added-power concept, we have introduced a fourth power gain $G_{\text{app}}$, called the apparent power gain, defined as the ratio between the output available power and the input power. We have investigated some properties of $G_{\text{app}}$ in the general and unilateral case, and shown how this gain can be exploited in the power transfer maximization. The analysis ultimately suggests that, despite its interesting mathematical properties, $G_{\text{app}}$ is indeed of little practical use in power transfer optimization.

**Author Contributions:** The two authors (G.G. and M.P.) contributed equally to the development of the project and to the preparation of the draft and revised versions of the paper. MATLAB scripts, simulations and the related figures where developed by G.G. All authors have read and agreed to the published version of the manuscript.

**Funding:** This research received no external funding.

**Data Availability Statement:** The MATLAB scripts used in the simulations are available from the authors under request. All data shown can be easily generated from the analytical formulae provided in the paper.

**Conflicts of Interest:** The authors declare no conflicts of interest.

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
