# Peer review of "Revisiting the Power Gains of a Loaded Two-Port: Is There a Missing Element?"

_electronics, doi:10.3390/electronics13030545_

Round 1

Reviewer 1 Report

Comments and Suggestions for Authors

The authors give a tutorial introduction to power gain definitions, and explain why the 4th type is not traditionally applied to power gain design. The article appears interesting to readers.

Reviewer 2 Report

Comments and Suggestions for Authors

This is an interesting, theoretical paper, which can also be of great interest for young microwave engineers to deeply understand the meaning of fundamental concepts as: power gain, conjugate matching, …

The novelty is quite limited, however previously discussed pros justify its publication in the present form.

The Reviewer would like to suggest giving a name to the new power gain. G4 is anonymous, also considering it is the central part of the manuscript. As a proposal, “Apparent Power Gain” can be a choice, justified by inequalities reported in the second paragraph and since it, as stated by the Authors, “does not refer to a physically realizable circuit configuration unless conjugate matching is imposed at both ports.”

Minor remarks

Paragraph 2: “A fourth possibility for defining again…” -> “A fourth possibility for defining a gain…”

If possible, describe more in detail Fig.2, since, in the Reviewer's opinion, it represents an added value for this manuscript.

Fig.2.c, Gav(GL) -> Gav(GG)
